# EBV Reactivation from Latency Is a Degrading Experience for the Host

**DOI:** 10.3390/v15030726

**Published:** 2023-03-11

**Authors:** Alejandro Casco, Eric Johannsen

**Affiliations:** 1Department of Oncology, McArdle Laboratory for Cancer Research, University of Wisconsin, Madison, WI 53705, USA; 2Department of Medicine, Division of Infectious Diseases, University of Wisconsin, Madison, WI 53705, USA

**Keywords:** herpesvirus, host shutoff, BGLF5, SOX, transcriptomics

## Abstract

During reactivation from latency, gammaherpesviruses radically restructure their host cell to produce virion particles. To achieve this and thwart cellular defenses, they induce rapid degradation of cytoplasmic mRNAs, suppressing host gene expression. In this article, we review mechanisms of shutoff by Epstein–Barr virus (EBV) and other gammaherpesviruses. In EBV, canonical host shutoff is accomplished through the action of the versatile BGLF5 nuclease expressed during lytic reactivation. We explore how BGLF5 induces mRNA degradation, the mechanisms by which specificity is achieved, and the consequences for host gene expression. We also consider non-canonical mechanisms of EBV-induced host shutoff. Finally, we summarize the limitations and barriers to accurate measurements of the EBV host shutoff phenomenon.

## 1. Introduction

Epstein–Barr virus (EBV) is a gammaherpesvirus that persistently infects more than 90% of the adult human population. Primary EBV infection often presents as infectious mononucleosis but may cause milder illness, particularly in children. After recovery, EBV infection is usually uneventful; however, the combined effects of high seroprevalence and life-long infection allow rare complications of chronic EBV infection to cause significant disease. In 2020, this included an estimated 300,000 EBV-associated human cancers, with approximately 200,000 cancer deaths [1]. EBV also causes nonmalignant diseases, including hemophagocytic lymphohistiocytosis and oral hairy leukoplakia, and has been linked to several auto-immune disorders, particularly multiple sclerosis [2,3].

Integral to EBV’s strategy of persistence is its ability to establish life-long latent infection in B lymphocytes and to reactivate from this reservoir to infect other hosts. A remarkable feature of this reactivation is that EBV produces virions asymptomatically leading to its presence in the saliva of seropositive individuals. In order to accomplish this maturation without inciting a host inflammatory response, EBV employs multiple mechanisms during lytic replication to co-opt cellular machinery and thwart innate and adaptive immunity [4]. Chief among these is EBV’s ability to induce widespread suppression of host gene expression once it commits to its early lytic cascade. This phenomenon is termed “host shutoff” and its effects on the host and EBV that permit successful replication are the focus of this review.

## 2. Host Shutoff Mechanisms Are Highly Divergent, Even among Herpesviruses

Although host shutoff is employed by many viruses from different viral families, the effector mechanisms vary widely. Within the herpesvirus family, the term “virion host shutoff (vhs)” was coined to describe the phenomenon of inhibition of cellular protein synthesis and host mRNA degradation upon infection of alphaherpesviruses such as herpes simplex virus (HSV) [5]. As the name implies, the protein mediating host shutoff (HS) in alphaherpesviruses is packaged within the virion and exerts its effects during initial infection. However, no homologs of the alphaherpesvirus vhs protein are encoded by beta or gammaherpesviruses (Table 1). Moreover, the betaherpesvirus family does not exhibit host shutoff during its replication [6]. A significant breakthrough in our understanding of host shutoff was the discovery that the DNA alkaline exonuclease (AE) found in all herpesviruses mediates host shutoff in gammaherpesviruses. Subsequent work demonstrated that in the gammaherpesvirus subfamily, this protein also has RNase activity, but unlike vhs, gammaherpes host shutoff factors (HSFs) are not packaged in virions [7,8,9,10]. Thus, in gammaherpesvirus infected cells, including EBV, host shutoff strictly occurs during reactivation from latency.

## 3. Host Shutoff in Epstein–Barr Virus Infection

Shortly after its discovery in KSHV, host shutoff was described for EBV by Rowe and colleagues, who used a CD2/eGFP lytic reporter gene to enrich Akata Burkitt lymphoma cells undergoing EBV lytic replication [11]. They observed a near-global inhibition of total de novo protein synthesis in these cells, including the synthesis of both HLA class I and II molecules. This effect occurred even when viral DNA synthesis was blocked, implicating an early lytic gene product in the effect. They further demonstrated that expression of BGLF5, the EBV homolog of KSHV ORF37 (also called SOX for ShutOff and eXonuclease), was sufficient to mediate host shutoff. Interestingly, both ORF37/SOX and BGLF5 appeared to exert their shutoff effects on mRNA stability [11] despite these proteins being AEs without known RNase activity.

Mechanistic studies of the gammaherpes HSFs BGLF5 and SOX initially suggested their shutoff function was independent of exonuclease activity. The activities were genetically separable: random mutagenesis screens produced BGLF5 and SOX mutants that were selectively impaired for either AE or HS activity [12,13,14]. The activities are also subcellularly separated. Processing of viral DNA by AE occurs exclusively in the nucleus and the HSV AE, which lacks HS function, localizes exclusively to the nucleus. By contrast, gammaherpesvirus AEs display nucleocytoplasmic localization [11,12,15]. This subcellular localization appears to play a role in regulating BGLF5/SOX function: AE activity in the nucleus and HS in the cytoplasm. Indeed, restricting SOX localization to the cytoplasm by abolishing its nuclear localization signal did not affect its HS activity [12]. Similarly, fusion of the murine gammaherpesvirus 68 SOX homolog (muSOX) to a nuclear retention signal restricted its localization to the nucleus and abolished HS activity [15]. Interestingly, structural studies established that purified BGLF5 and SOX have intrinsic RNase activity that depends on the same catalytic machinery as AE DNA processing [16,17]. Thus, the genetic separability of AE and HS may be due to the involvement of different residues in substrate recognition (and/or subcellular localization) rather than separability of activities per se. In a landmark study, Covarrubias et al. demonstrated that SOX degrades mRNAs via a two-step process: The initial endonucleolytic cleavage is mediated by SOX, making the two fragments susceptible to degradation by cellular exonucleases (Figure 1A) [18,19]. This two-step mechanism was subsequently shown to be conserved across other host shutoff factors, including BGLF5 [20]. This mechanism contrasts with basal cellular mRNA decay that initiates at mRNA ends via poly(A) removal and decapping. The internal cleavage of target mRNAs by BGLF5 and homologs is hypothesized to have a two-fold effect: (1) rapid inactivation of transcripts rendering them incompetent for translation and (2) a mechanism for transcript selectivity. However, the extent of selectivity is incompletely defined.

## 4. Selectivity

Several early studies indicated that BGLF5 and its homologs have target specificity despite inducing widespread RNA degradation. For example, RNAs transcribed by polymerase II (RNAPII) are degraded, but noncoding RNAs and even untranslatable mRNAs are not. This specificity is thought to arise through targeting of mRNAs associated with the translation machinery [17,19,20,21]. Indeed, in vitro studies have suggested that BGLF5 and SOX association with RNA is substantially weaker than with DNA and may require additional factors in vivo to efficiently target and degrade RNA [16,17]. Consistent with this suggestion, polysome profiling revealed that SOX co-sediments with the 40S ribosomal subunit [19,20], which would allow preferential targeting of mRNAs and may be important for enhancing binding to RNA and/or the efficiency of cleavage. Future studies examining whether BGLF5 also associates with the 40S subunit and the extent to which BGLF5/SOX mutants selectively defective for HS activity are impaired for 40S association will help clarify the significance of this proposed mechanism for selectivity.

Gammaherpesvirus HSFs also appear to mediate site-specific cleavage of their target transcripts. For example, primary cleavage of reporter mRNAs by HSF (prior to exonuclease degradation) produces fragments of consistent size [19,20]. Furthermore, primary cleavage of different transcripts of similar length results in distinct cleavage intermediates [19]. In addition, cleavage of the same transcript by BGLF5 versus SOX resulted in different intermediates [20]. Thus, despite their shared mechanism, gammaherpesvirus HSFs likely recognize different target sequences in host mRNAs and therefore may preferentially target distinct populations of host mRNAs. Characterization of cleavage intermediates produced by SOX from three reporter mRNAs identified a five-base motif (UGAAG) upstream of the cleavage site that was necessary but not sufficient for SOX cleavage [19]. In contrast, a larger 200nt sequence containing this motif did confer susceptibility to SOX cleavage. Given that the 200nt sequences from 3 reporter mRNAs were different, this finding supports a requirement for specific mRNA structure(s) in addition to the cis acting motif. A transcriptome-wide analysis of SOX cleavage was also consistent with site-specific cleavage [22]. Although no simple consensus sequence emerged, sites could be predicted by a position-specific weight matrix. RNA structure prediction indicated susceptible sequences are characterized by a secondary RNA stem-loop that is frequently followed by several unpaired adenine residues just upstream of the cleavage site. Mendez et al. demonstrated SOX binds specifically to this polyadenosine tract and that both binding and cleavage require an open loop structure [23]. Further in vitro and in silico studies, including the solved structure of SOX bound to a target RNA, confirmed that cleavage occurs within unpaired nucleotides found within a stem-loop or bulge-loop structure [24]. These authors also found that SOX makes no sequence specific contacts with RNA. Thus, the four adenine residues upstream of the cleavage site, by virtue of their decreased propensity to form RNA duplexes, are promoting loop structures, not sequence specific binding. The SOX residues responsible for binding these RNA-loop structures are only partially conserved in BGLF5. This difference in the binding pocket provides additional evidence that gammaherpesvirus HSFs target different, but potentially partially overlapping, sets of host mRNAs. Collectively, these studies have demonstrated that the gammaherpesvirus host shutoff factors target mRNAs at an early stage during translation and that cleavage occurs at specific sites determined largely by RNA structural motifs. Such motifs have been shown to be widely distributed across host and viral RNAs [22], explaining the ability of these factors to cause widespread degradation while selectivity targeting mRNAs competent for translation.

## 5. Host Shutoff Escape

A surprising feature of herpesvirus HSFs is that they also target viral mRNAs for degradation [13,21,24,25,26]. HSF cleavage activity needs to be sufficiently limiting, at least at physiologic levels of BGLF5 or SOX expression, to ensure adequate viral protein levels are obtained from the surviving viral mRNAs. In addition, some host mRNAs are resistant to HSF cleavage. The first such resistant mRNA identified was IL-6, which accumulates at high levels in KSHV infected cells undergoing lytic replication despite SOX expression [27]. Deep sequencing of mRNAs from HEK293 cells identified multiple other transcripts that were not decreased upon SOX expression. Some, like the apoptosis enhancing nuclease (AEN) transcript, were directly resistant to cleavage, while others were susceptible to cleavage, but did not decrease in abundance, presumably due to other regulatory effects such as increased transcription. The 3′ UTRs of the IL-6, GADD45B, and C19ORF66 transcripts contain elements that are sufficient to mediate resistance to HSF cleavage when transferred to a heterologous reporter mRNA [28,29,30]. Some of these SOX-resistance elements (SREs) also mediate BGLF5 resistance. In contrast, SREs found in the AEN and GADD45B transcripts failed to confer resistance to muSOX or BGLF5 cleavage [29,31].

The mechanism by which these SREs confer resistance is incompletely understood. Several studies implicated host ribonucleoprotein complexes binding to SREs as being essential for their role in mediating HSF resistance [28,29,32,33]. SREs from different transcripts appear to bind distinct but partially overlapping sets of RNPs. In several cases, siRNA knockdown of components of these SRE-binding RNP complexes restored susceptibility to HSF degradation. Whether differences in the RNP complexes account for the ability of some SREs to confer resistance to cleavage by multiple HSFs and other SREs to confer resistance to a single HSF is unclear. Despite the uncertainty regarding the exact mechanism by which SREs act, it is clear that differential transcript resistance and susceptibility to specific HSF proteins combine to define the spectrum of host mRNAs targeted for degradation during replication of a given gammaherpesvirus. It is likely that some host mRNAs (as occurs with viral mRNAs—see Section 8) are expressed as sufficiently high to be translated into their protein products despite susceptibility to HSF degradation. Furthermore, because HSF acts at the mRNA level, many host proteins persist despite degradation of their corresponding mRNAs. Therefore, it is predominantly new protein expression that is affected [11,34].

## 6. Secondary HSF Effects May Augment Shutoff

Extensive degradation of mRNAs by gammaherpesvirus HSFs may further augment shutoff of host gene expression by dysregulating mRNA stability factors. BGLF5 or SOX expression leads to marked redistribution of RNA binding proteins including the cytoplasmic poly(A)-binding protein (PABPC) into the nucleus as well as hyperadenylation of nascent mRNAs [25,35,36]. PABPC redistribution is downstream of the RNA degradation activity of HSFs, presumably due to release of large amounts of PABPC from degraded host mRNAs [37]. Nuclear accumulation of PABPC may impair mRNA export and be responsible for mRNA hyperadenylation induced by HSFs [35]. Remarkably, this HSF-induced RNA degradation is accompanied by decreased RNAPII recruitment to host promoters by an unknown mechanism, leading to repressed transcription of many host genes [18]. While not formally demonstrated to occur with BGLF5, this degradation-induced transcriptional repression is seen with SOX and HSV vhs, suggesting it is a general feature of herpesvirus host shutoff. Notably, transcription of viral mRNAs does not occur via this mechanism. As these secondary host shutoff mechanisms only occur as a consequence of HSF-induced mRNA degradation, it is not clear that their contributions are essential for achieving shutoff. However, these secondary effects do provide a means by which HSF can impair expression of mRNAs that resist their endonuclease activity.

## 7. Host Genes Targeted by HSF during Lytic Infection

Host shutoff may contribute to viral replication by at least two mechanisms: abrogating protein expression required to mount an anti-viral response and promoting preferential translation of viral mRNAs. There is substantial evidence that EBV host shutoff can impair the innate and adaptive immune responses. EBV lytic reactivation in Burkitt lymphoma cells is associated with downregulation of multiple Toll-like receptors, including TLR1, TLR6, TLR7, TLR9, and TLR10. Overexpression of BGLF5 inhibits TLR2 and TLR9 expression in HEK293 or MelJuSo melanoma cells [38,39]. Both TLR2 and TLR9 are implicated in sensing and inhibiting primary EBV infection and reactivation [40,41,42,43,44]. BGLF5 also targets CD1d, a non-classical HLA that presents lipid antigens to invariant natural killer T (iNKT) cells [38,45]. The iNKT response has been implicated in control of primary EBV infection and may play a role in limiting EBV reactivation during chronic infection [46,47]. In contrast, TLR4 which has not been reported to sense EBV infection, was not downregulated by BGLF5 overexpression, indicating some degree of selectivity of innate immune factors targeted by BGLF5 [38,42].

BGLF5 is capable of targeting genes essential for adaptive immune responses, including HLA class I and II molecules [11,14,25,38]. Further, expression of BGLF5 or SOX protects HEK293 cells against HLA-A2-restricted CD8^+^ T-cell responses compared to control [14]. This protection was conferred by BGLF5 mutants defective for alkaline exonuclease but not BGLF5 lacking host shutoff activity—providing a direct link between host shutoff and immune evasion. Evidence that BGLF5 protects EBV-infected B cells from adaptive immune responses is less clear. Knockdown of BGLF5 in lymphoblastoid cell lines (LCLs) only produced a slight increase in CD8^+^ T-cell response to specific lytic antigens compared to knockdown or knockout of other EBV immune evasion genes (BILF1 or BNLF2a) [48]. This discrepancy highlights the potential redundancy of EBV anti-immune factors; however, it is at least noteworthy that BGLF5 is not sufficient to impair the CD8^+^ T-cell response (i.e., in the BILF1 and BNLF2a knockdown conditions). It should also be noted that these CD8+ T-cell responses were maintained despite decreased lytic protein expression observed during the knockdown of BGLF5. One important caveat of these studies is the BGLF4 and BGLF5 transcripts overlap such that any shRNA that targets BGLF5 will also knockdown BGLF4 transcript and significant functional interactions between BGLF4 and BGLF5 have been reported [26,49,50]. Nonetheless, BGLF5 does abrogate adaptive immune responses, though further studies in relevant cell types are required to define the magnitude and extent of this effect.

## 8. HSF Interactions with Viral Gene Expression

It is unlikely that HSF promotes preferential translation of viral mRNAs. Multiple studies have demonstrated that viral mRNAs are susceptible to HSFs, an effect that is presumably overcome by high-level transcription of lytic mRNAs [13,21,24,25,26]. Thus, HSFs may fine tune expression levels of viral mRNAs and HSF knockout has the potential to disrupt the stoichiometry of viral proteins required for optimal viral replication. Although knockdown of BGLF5 impairs nucleocapsid maturation and slightly impairs DNA replication, and the former may be due to loss of AE rather than host shutoff activity [51]. The situation is clearer for MHV68, where a host shutoff-specific muSOX mutant (R443I) produces virions with abnormal morphology and composition [21]. Interestingly, the R443I mutant did not exhibit noticeable replication defects during acute infection in vivo but did impair the establishment of latency as evidenced by reduced numbers of infected splenocytes and lower levels of viral DNA during chronic infection [13].

Viral genes may in turn regulate HSF activity. This possibility is best established for herpes simplex virus, where its vhs is inactivated at late stages of replication to facilitate accumulation of late mRNAs. It is not known whether this inactivation occurs for gammaherpesvirus HSFs. For EBV, it has been suggested that the BGLF4 protein kinase counters BGLF5 shutoff based on their opposing effects on several viral mRNAs in transcomplementation assays [26]. However, evidence is conflicting regarding whether BGLF4 promotes BGLF5 phosphorylation [52,53]. Furthermore, when the number of cells in the late phase is taken into account, BGLF5-knockout produces subtle increases in late mRNA abundance [26,51,54]. One important limitation of these studies is that they were performed with bulk populations of asynchronously replicating cells. It is therefore difficult to isolate events occurring during the late phase of replication.

## 9. Non-HSF Viral Proteins Contribute to Impaired Host Gene Expression

Although canonical host shutoff by gammaherpes HSFs is sufficient to dramatically curtail host gene expression, several other viral proteins create additional “non-canonical shutoff” barriers that thwart expression of host genes (Figure 1B–D). For example, Buschle et al. recently demonstrated that the immediate early transcription factor Zta, in addition to initiating the EBV lytic cascade, induces a global restructuring of host chromatin with widespread loss of chromatin accessibility and chromatin–chromatin interactions [55]. They did not observe a decrease in host mRNA levels after 6 h of Zta expression, but this may not have been sufficient time for steady state expression levels to be achieved. Nascent transcription has been examined in Burkitt lymphoma cells induced for EBV replication with sodium butyrate. Reduction in transcription of multiple host genes was observed, potentially due to Zta effects and/or secondary effects downstream of sodium butyrate [56]. Further, Park et al. reported decreased synthesis of some host proteins in response to Zta expression [36]. Additionally, herpesvirus replication is characterized by a shift from preferential export of spliced mRNAs to preferential export non-spliced viral mRNAs. In EBV this change is mediated by SM (also known as Mta or EB2) and appears to be important for both export of viral mRNAs and their translation [57,58,59,60,61,62,63]. Progression of the lytic cascade is accompanied by such extensive compaction and margination of host chromatin that further host transcription becomes untenable [64,65]. These additional mechanisms may contribute to the ability of herpesviruses to usurp their host’s translational machinery. Additionally, any effort to define BGLF5 targets under physiologic conditions must be able to distinguish bona fide host shutoff effects from those due the potentially confounding host shutoff mechanisms.

## 10. Technical Barriers to Defining HSF Effects

Despite remarkable progress in understanding the mechanism(s) by which gammaherpesviruses induce host shutoff, many important questions remain. Because gammaherpesviruses are notoriously difficult to induce into lytic replication, many studies have relied upon HSF overexpression to define their effects. We know that HSF degradation is surmountable by viral mRNAs and likely by host mRNAs as well. Thus, overexpression-based approaches may exaggerate the true extent of host shutoff and will not capture regulatory effects of other viral genes on HSF activity. Enrichment strategies to isolate lytically infected cells can be used to circumvent these limitations (Figure 2). One such strategy is the use of reporter cell surface markers in conjunction with magnetic-activated cell sorting (MACS) [11,39,55,66,67]. However, several limitations of MACS make it less suitable for enriching lytic reactivated cells. First, the enrichment efficiency of MACS is inversely related to the rarity of the target population, with increasing rarity resulting in decreased efficiency of enrichment [68,69]. Second, the asynchronous nature of lytic reactivation produces distinct early and late lytic sub-populations that require separate enrichment. Alternatively, single cell RNA-seq approaches could be employed, although the low frequency of lytic infection may make the cost of this approach prohibitive.

Another technical challenge in studying host shutoff is that most RNA quantification methods, especially RNA-seq, make the implicit assumption that the total amount of mRNA in the cell is the same among different experimental conditions. This assumption is clearly false because of the extensive mRNA degradation resulting from host shutoff. Failure to account for this problem is highlighted by two studies analyzing similar lytic transcriptomes of Akata Burkitt lymphoma cells using different normalization methods. Conventional RNA-seq normalization by Ramasubramanyan et al. [66] indicated few changes in host gene expression during lytic replication. In contrast, by using synthetic spike-in RNA standards for their normalization, Buschle et al. [55] accounted for differences in transcript abundances across lytic and non-lytic conditions to demonstrate a widespread host shutoff resulting in a massive downregulation of host genes. It is also important to consider that general RNA-seq analysis may be difficult in the context of host shutoff. For example, gene set enrichment analysis may be less helpful in identifying meaningful biological pathways important for viral replication given that the overall imprint is suppression of host pathways. Alternatively, analyzing unregulated genes (i.e., host shutoff escapees) may be more useful. Although this approach is less conventional than analysis of differentially expressed genes, commonly used tools including DESeq2 incorporate statistical methods for detecting unregulated genes [70]. Such considerations are necessary for accurate and meaningful analysis of lytic replication in the face of host shutoff.

Studying lytic replication in physiologically normal cells is also important for determining biologically relevant host shutoff targets and escapees. Differences in transcripts targeted or escaping HSF-mediated degradation across cell types has been previously described [28,29,30,31]. These differences are due in part to expression of lineage specific mRNAs but may also be due to differences in HSF expression levels, the presence/absence of proteins, or mRNA isoforms that promote host shutoff escape. To date, all studies have relied on non-physiologic cell lines due to limitations in cell culture described earlier. For example, cell lines derived from Burkitt lymphomas are commonly used due to their higher levels of lytic replication [55,56,66,71,72]. Because HSF are under selection for their ability to allow gammaherpes viruses to replicate in normal cells, it is essential that HSF effects be defined in model systems that most closely approximate them. Studies using physiologic models such as lymphoblastoid cell lines and EBV-infected oral keratinocytes to define the effects of BGLF5 during EBV replication form an important basis for defining the essential targets of EBV shutoff.

## Figures and Tables

**Figure 1 viruses-15-00726-f001:**
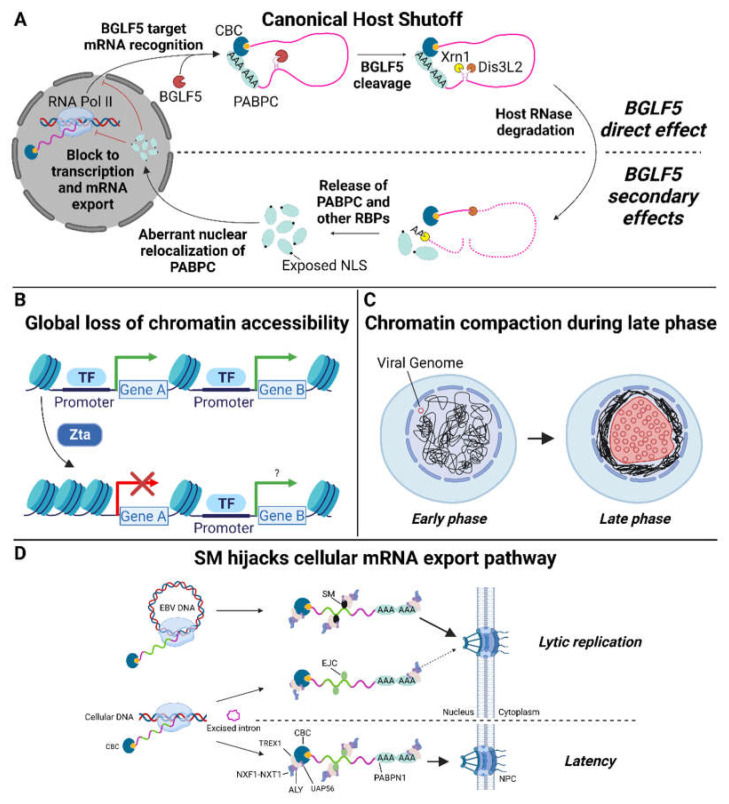
Canonical and non-canonical mechanisms of EBV-mediated host shutoff. (**A**) Canonical EBV host shutoff occurs in the cytoplasm via a two-step process: initial endonucleolytic cleavage by BGLF5 and subsequent degradation by cellular exonucleases (Xrn1, Dis3L2, and likely the exosome complex (not shown)). mRNA degradation releases RNA binding proteins (RBPs), which are recycled into the nucleus. The accumulation of RBPs, in particular the cytoplasmic poly(A)-binding protein (PABPC), induces a state of cellular stress that inhibits host gene expression by inhibiting transcription by RNA polymerase II and possibly by suppressing the nuclear export of mRNAs. (**B**–**D**) Non-canonical EBV host shutoff mechanisms include: (**B**) global loss of host chromatin accessibility induced by the immediate early protein Zta. (**C**) During the late phase of replication, host chromatin compaction occurs in parallel with the expansion of viral replication compartments that is believed to further disrupt the ability of the host cell to express its gene repertoire. (**D**) Preferential export of non-spliced viral mRNA mediated by the EBV SM protein. Abbreviations: NLS, nuclear localization signals; CBC, cap-binding complex; TF, transcription factor. Figure created with BioRender.com.

**Figure 2 viruses-15-00726-f002:**
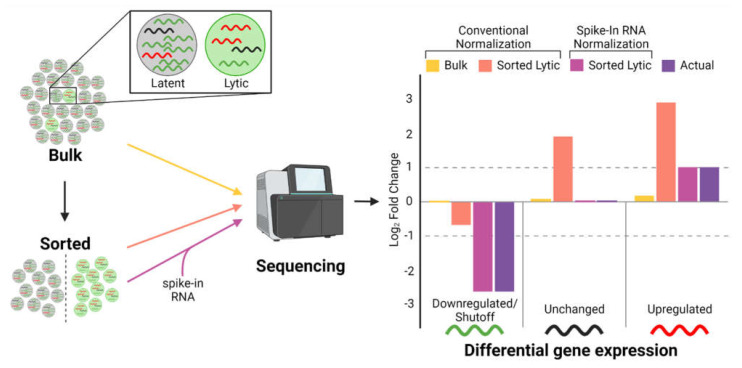
Barriers and potential solutions for accurate quantification of host shutoff. The inefficient and asynchronous entry of EBV into lytic replication means that bulk populations of EBV infected cells are predominantly latently infected with decreasing levels of early and late lytic infection. As a consequence, detection of differentially expressed genes is diminished, especially mRNAs that are downregulated (or degraded) in lytic cells, such as those subject to host shutoff (compare bulk to actual). This limitation can be addressed via cell sorting approaches; however, given the marked differences in total mRNA in latent versus lytic cells, conventional normalization methods fail to accurately measure changes in gene expression. To account for such differences, exogenous spike-in RNAs can be added on a per cell basis. These (or similar) modifications are essential to capture host shutoff effects accurately on the cell gene expression. Figure created with BioRender.com.

**Table 1 viruses-15-00726-t001:** Diverse herpesvirus genes mediate host shutoff. Virion host shutoff (vhs) protein homologs are found only in alphaherpesviruses. In gammaherpesviruses, host shutoff is mediated by the alkaline exonuclease (AE) proteins. Collectively, herpesvirus proteins that promote degradation of host mRNAs are referred to as host shutoff factors (HSF) and are shaded green. Note that host shutoff is not observed during betaherpesvirus replication.

	virus	VHS	AE
alpha	HSV	UL41	UL12
	VZV	ORF17	ORF48
beta	CMV	–	UL98
	HHV6	–	U70
	HHV7	–	U70
gamma	EBV	–	BGLF5
	KSHV	–	ORF37/SOX

## Data Availability

Not applicable.

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
