# Peer review of "EBV Reactivation from Latency Is a Degrading Experience for the Host"

_viruses, 2023, doi:10.3390/v15030726_

Round 1

Reviewer 1 Report

The review by Alejandro C and Eric J summarizes the role of host shutoff protein during EBV’s reactivation from latency to the lytic phase. This manuscript also discusses the possibility of leveraging HSF to resolve the scientific challenge to induce EBV into the lytic replication cycle. This manuscript is well written; however, I have a few minor comments below:

1.     Line 219: SOX, BGLF5’s homolog of in KSHV has similar functions in the majority of the host shutoff mechanism, does SOX also show similar abrogate adaptive immune responses?

2.     Is there any literature demonstrating whether BGLF5 is packaged into EBV virion?

Author Response

The review by Alejandro C and Eric J summarizes the role of host shutoff protein during EBV’s reactivation from latency to the lytic phase. This manuscript also discusses the possibility of leveraging HSF to resolve the scientific challenge to induce EBV into the lytic replication cycle. This manuscript is well written; however, I have a few minor comments below:

1. Line 219: SOX, BGLF5’s homolog of in KSHV has similar functions in the majority of the host shutoff mechanism, does SOX also show similar abrogate adaptive immune responses?
Response: We know less about SOX’s effect on the immune response, but had added information regarding this where possible.

2. Is there any literature demonstrating whether BGLF5 is packaged into EBV virion?
Response: We covered this on lines 51-53 where we state that BGLF5 and the other gammaherpes HSFs have NOT been found in virions.

Reviewer 2 Report

This review by Casco and Johannsen elegantly summarizes key aspects of the gamma-herpesvirus host shutoff field.  It briefly covers alpha-herpesvirus shutoff and then focuses on KSHV and EBV host shutoff proteins. It successfully covers a lot of territory, including molecular biology of gHHV HS target host transcripts, effects of host shutoff on host immune responses, and other mechanisms by which EBV perturbs host gene expression in the lytic cycle, including at the level of chromatin fragmentation and nuclear RNA export.  Overall, it is well written and the Biorender figures are a nice addition. It will be a fitting contribution to the field and I only have minor suggestions on its way to publication:

The subject of RNAs that escape BGLF5 comes up in two places. Early in the text there is a focus on host mRNA elements that facilitate escape.  Why might gHHV have evolved to avoid degrading these transcripts?  Later in the text, the subject of viral mRNA escape comes up, with the notion that volume allows escape.  I would suggest discussing these together, rather than in two separate pages.  It does raise the question of whether abundant host transcripts likewise evade shutoff by analogy with herpesviral transcripts.  Are herpesvirus transcripts actually more abundant than host transcripts such as GAPDH?

Would clarify this statement in line 80: “have intrinsic RNase activity that depends on the same catalytic machinery as DNA processing”. What type of DNA processing?  Not sure what this refers to.

Why might beta HHV have evolved to apparently lose HS, given its presence in alpha and gamma HV? Would speculate on why CMV’s AE doesn’t have HS function, despite it’s ability to infect hosts and reactivate in a fairly similar compartment as EBV/KHSV.  Given the impressive extent to which CMV has evolved a battery of immune evasion genes, why might it have lost HS? What can be gleaned from its expression pattern, structure, etc?  Is it retained in the nucleus? 

Some of the text in fig 1 is too small to read unless the figure is blown way up, such as the AAA sequences bound by PABC. Likewise, in 1C, ‘viral genome’ font is too small. Same for labels in 1D. would try to keep a font like Arial 9 throughout.  Since space is not a limitation, would advise that all figures should be increased in size, so that they can be better appreciated when printed out. Likewise, Section 9 should provide a call out to Figure 1. However, I would also suggest breaking fig 1 into two figures, and having the sections on ZTA and SM be placed as a figure 2 to accompany section 9, where this comes up on in the text.

It would be a very nice addition to discuss that proteome level analysis of B cells triggered for lytic replication has been performed, since this gets at an aspect that is missing in the review – how much does host shutoff ultimately affect the proteome of lytic cells, which is the payoff of HS. In PMID 28514666, plasma membrane proteome analysis was done on samples normalized for cell number.  A broad shutoff effect on host protein expression was not seen at the plasma membrane out to 72 hours post induction of lytic replication. For whole cell analysis, despite the caveat that protein levels were normalized across timepoints, this did not result in apparent increased expression of host proteins encoded by transcripts that escape BGLF5 – they should have appeared over-expressed if the analysis falsely increased protein expression across the board by normalizing protein at 0, 48 and 72 hrs of lytic replication, as they would have a higher relative expression.  Surely gamma-herpesviruses depend on a large fraction of the proteome remaining in tact in order to be able to synthesize, traffic and export virion from live cells, which require a wide swath of cell biology and metabolism to remain in tact.  Similarly, TLRs are discussed as a target for VHS. But, TLRs are abundantly expressed in membranes of host cells when they initiate lytic replication. It is unclear to me that HS would have a window to perturb TLR responses on the protein level in a way that would impact a TLR response – they are preformed and ready to signal.  And since TLR9 senses methylated CpG,  what is the relevance to lytic infection, particularly if HS blocks downstream transcriptional responses? 

Additional text corrections:

Line 47: there are no homologs to the alphaherpesvirus vhs protein -> no homologs of the alphaherpesvirus..

Line 53-4: Thus,  in gammaherpesviruses -> Thus, in gammaherpesvirus infected cells,

Line 57: Host shutoff in Epstein-Barr virus -> Host shutoff in Epstein-Barr virus infection.

66-7: despite these proteins being an AE without known RNase activity-> despite these proteins being AEs without known RNase activity

Line 71, clarify that HS is host shutoff. The abbreviation isn’t defined.

145-7 a bit tough to read. Would rephrase, add commas, etc.  Also, For should be spelled four.s These authors also found that SOX makes no sequence specific contacts with RNA consistent with the for adenine residues upstream of the cleavage site reflecting their ability 146 to distort RNA duplexes (i.e., promote unpaired structures

Line 313 typo: Failure to account for problem this is highlighted

Author Response

This review by Casco and Johannsen elegantly summarizes key aspects of the gamma-herpesvirus host shutoff field.  It briefly covers alpha-herpesvirus shutoff and then focuses on KSHV and EBV host shutoff proteins. It successfully covers a lot of territory, including molecular biology of gHHV HS target host transcripts, effects of host shutoff on host immune responses, and other mechanisms by which EBV perturbs host gene expression in the lytic cycle, including at the level of chromatin fragmentation and nuclear RNA export.  Overall, it is well written and the Biorender figures are a nice addition. It will be a fitting contribution to the field and I only have minor suggestions on its way to publication:

The subject of RNAs that escape BGLF5 comes up in two places. Early in the text there is a focus on host mRNA elements that facilitate escape.  Why might gHHV have evolved to avoid degrading these transcripts?  Later in the text, the subject of viral mRNA escape comes up, with the notion that volume allows escape.  I would suggest discussing these together, rather than in two separate pages.  It does raise the question of whether abundant host transcripts likewise evade shutoff by analogy with herpesviral transcripts.  Are herpesvirus transcripts actually more abundant than host transcripts such as GAPDH?

Response:  We agree that highly expressed host mRNAs likely surmount HSF degradation and now mention this in our discussion of escape.  We make no claims that the gHHV have evolved to allow these mechanisms of escape.  It is entirely possible that this phenomenon is evolutionarily neutral

Would clarify this statement in line 80: “have intrinsic RNase activity that depends on the same catalytic machinery as DNA processing”. What type of DNA processing?  Not sure what this refers to.

Response: we now clarify that this refers to the AE activity.

Why might beta HHV have evolved to apparently lose HS, given its presence in alpha and gamma HV? Would speculate on why CMV’s AE doesn’t have HS function, despite it’s ability to infect hosts and reactivate in a fairly similar compartment as EBV/KHSV.  Given the impressive extent to which CMV has evolved a battery of immune evasion genes, why might it have lost HS? What can be gleaned from its expression pattern, structure, etc?  Is it retained in the nucleus?

Response:  There are interesting questions, but beyond the scope of this review.  One could speculate that the battery  of immune evasion genes expressed by CMV obviates the need for HS.

Some of the text in fig 1 is too small to read unless the figure is blown way up, such as the AAA sequences bound by PABC. Likewise, in 1C, ‘viral genome’ font is too small. Same for labels in 1D. would try to keep a font like Arial 9 throughout.  Since space is not a limitation, would advise that all figures should be increased in size, so that they can be better appreciated when printed out. Likewise, Section 9 should provide a call out to Figure 1. However, I would also suggest breaking fig 1 into two figures, and having the sections on ZTA and SM be placed as a figure 2 to accompany section 9, where this comes up on in the text.

Response:  We have modified the figure to make the text more easily readable and added a call out to Figure 1 in section 9 as suggested.

It would be a very nice addition to discuss that proteome level analysis of B cells triggered for lytic replication has been performed, since this gets at an aspect that is missing in the review – how much does host shutoff ultimately affect the proteome of lytic cells, which is the payoff of HS. In PMID 28514666, plasma membrane proteome analysis was done on samples normalized for cell number.  A broad shutoff effect on host protein expression was not seen at the plasma membrane out to 72 hours post induction of lytic replication. For whole cell analysis, despite the caveat that protein levels were normalized across timepoints, this did not result in apparent increased expression of host proteins encoded by transcripts that escape BGLF5 – they should have appeared over-expressed if the analysis falsely increased protein expression across the board by normalizing protein at 0, 48 and 72 hrs of lytic replication, as they would have a higher relative expression.  Surely gamma-herpesviruses depend on a large fraction of the proteome remaining in tact in order to be able to synthesize, traffic and export virion from live cells, which require a wide swath of cell biology and metabolism to remain in tact.  Similarly, TLRs are discussed as a target for VHS. But, TLRs are abundantly expressed in membranes of host cells when they initiate lytic replication. It is unclear to me that HS would have a window to perturb TLR responses on the protein level in a way that would impact a TLR response – they are preformed and ready to signal.  And since TLR9 senses methylated CpG,  what is the relevance to lytic infection, particularly if HS blocks downstream transcriptional responses?

Response:  We had cited PMID 28514666  in the review as reference 34; however, we inadvertently omitted this publication from the reference list.  Thank you for catching this errors as it was  not only unfortunate that we appeared not to have discussed this, but all of our references above #34 were misnumbered as a result of this omission.  We have fixed both problems in this revision.

Additional text corrections:

Line 47: there are no homologs to the alphaherpesvirus vhs protein -> no homologs of the alphaherpesvirus.

Line 53-4: Thus,  in gammaherpesviruses -> Thus, in gammaherpesvirus infected cells,

Line 57: Host shutoff in Epstein-Barr virus -> Host shutoff in Epstein-Barr virus infection.

66-7: despite these proteins being an AE without known RNase activity-> despite these proteins being AEs without known RNase activity

Line 71, clarify that HS is host shutoff. The abbreviation isn’t defined.

145-7 a bit tough to read. Would rephrase, add commas, etc.  Also, For should be spelled four.s These authors also found that SOX makes no sequence specific contacts with RNA consistent with the for adenine residues upstream of the cleavage site reflecting their ability 146 to distort RNA duplexes (i.e., promote unpaired structures

Line 313 typo: Failure to account for problem this is highlighted

Response:  We appreciate each of these edits and have made appropriate changes for all in our revised manuscript.

Reviewer 3 Report

The review article by Alejandro Casco and Eric Johannsen discusses the host shutoff mechanisms in EBV and other herpesviruses. The topic represents an important element in the biology of these viruses that still requires further investigation. The review addresses differences among herpesvirus proteins that mediate host shutoff and the responsible domains. EBV and KSHV encode host shutoff factors BGLF5 and SOX, respectively. A two-step mechanism was discussed for the SOX host shutoff protein in KSHV that involves SOX-mediated endonucleolytic activity followed by host-mediated exonucleolytic activity of host and viral mRNAs. Additionally, the review summarizes how cleavage of mRNAs occurs at specific sites that seem to have unique structural motifs. A select group of host transcripts and many viral mRNAs escape the shutoff process. The article describes potential escape mechanisms that include upregulation of the transcription rate and the presence of elements that resist host shutoff cleavage. The exact mechanism by which RNA motifs grant resistance against degradation by host shutoff factors is still unclear. Interestingly however, SOX-resistant elements failed to protect against BGLF5-mediated cleavage. The significance of the host shutoff process in promoting viral replication was discussed, especially its importance in disrupting innate anti-viral responses that could interfere with lytic viral gene expression and production of new virus particles. It is possible that host shutoff might serve as a mechanism for turning off certain viral processes at the different stages of the lytic cascade, however; this point was not adequately discussed. This idea is supported by reports demonstrating that viral mRNAs are subject to cleavage by host shutoff factors. The concept that the process of host shutoff is regulated by other viral proteins is interesting but needs further investigation. Collectively, the article is well written and nicely summarizes old and recent findings in this field.

Additional remarks:

Perhaps the authors should consider adding a diagram comparing the different domains in alpha, beta and gamma herpesvirus proteins that mediate host shutoff.

The finding by Park et al demonstrating the potential role of EBV BZLF1 protein in directly mediating host shutoff was not discussed.

Lines 101-103 suggests that nascent host mRNA transcription has not been evaluated. Please refer to the report by Frey et al., 2020 Journal of Virology vol 94, Nascent Transcriptomics Reveal Cellular Prolytic Factors Upregulated Upstream of the Latent-to-Lytic Switch Protein of EBV. Also, the same report demonstrated increased host RNA synthesis for a subset of genes to counter EBV-induced host shutoff but was not cited for this important finding (lines 164-167).

Line 148-150. This difference in the binding pocket provides additional evidence that gammaherpesvirus host shutoff factors target different host mRNAs. It is likely that there are subsets of host and viral genes that are targeted by both BGLF5 and SOX.  

Line 93, rendering them incompetent for translation, rather than rendering them for incompetent translation.

Line 146, consistent with the “for”, for is misplaced.

Author Response

The review article by Alejandro Casco and Eric Johannsen discusses the host shutoff mechanisms in EBV and other herpesviruses. The topic represents an important element in the biology of these viruses that still requires further investigation. The review addresses differences among herpesvirus proteins that mediate host shutoff and the responsible domains. EBV and KSHV encode host shutoff factors BGLF5 and SOX, respectively. A two-step mechanism was discussed for the SOX host shutoff protein in KSHV that involves SOX-mediated endonucleolytic activity followed by host-mediated exonucleolytic activity of host and viral mRNAs. Additionally, the review summarizes how cleavage of mRNAs occurs at specific sites that seem to have unique structural motifs. A select group of host transcripts and many viral mRNAs escape the shutoff process. The article describes potential escape mechanisms that include upregulation of the transcription rate and the presence of elements that resist host shutoff cleavage. The exact mechanism by which RNA motifs grant resistance against degradation by host shutoff factors is still unclear. Interestingly however, SOX-resistant elements failed to protect against BGLF5-mediated cleavage. The significance of the host shutoff process in promoting viral replication was discussed, especially its importance in disrupting innate anti-viral responses that could interfere with lytic viral gene expression and production of new virus particles. It is possible that host shutoff might serve as a mechanism for turning off certain viral processes at the different stages of the lytic cascade, however; this point was not adequately discussed. This idea is supported by reports demonstrating that viral mRNAs are subject to cleavage by host shutoff factors. The concept that the process of host shutoff is regulated by other viral proteins is interesting but needs further investigation. Collectively, the article is well written and nicely summarizes old and recent findings in this field.
Response:   Thank  you.

Additional remarks:
Perhaps the authors should consider adding a diagram comparing the different domains in alpha, beta and gamma herpesvirus proteins that mediate host shutoff.
Response:  It was our intent that Table 1 would serve as a comparison among these families.  It highlights that beta herpesviruses lack any proteins that mediate host shutoff and that there is no homology between the proteins mediating host shutoff between alphas and gammas.  

The finding by Park et al demonstrating the potential role of EBV BZLF1 protein in directly mediating host shutoff was not discussed.
Response: We did discuss the direct effect of BZLF1 on PABPC relocalization.  It is our opinion that  the data in this paper do not distinguish between canonical and non-canonical mechanisms for the effects of BZLF1 on host proteins described.  The former would be unprecedented (and unlikely).  We therefore have added the Park reference to our discussion of BZLF1’s non-canonical host shutoff effects (lines 295-296).

Lines 101-103 suggests that nascent host mRNA transcription has not been evaluated. Please refer to the report by Frey et al., 2020 Journal of Virology vol 94, Nascent Transcriptomics Reveal Cellular Prolytic Factors Upregulated Upstream of the Latent-to-Lytic Switch Protein of EBV. Also, the same report demonstrated increased host RNA synthesis for a subset of genes to counter EBV-induced host shutoff but was not cited for this important finding (lines 164-167).
Response:  We agree that it was an unfortunate oversight not to discuss this manuscript.  In the context cited 101-103, it is correct to say that nascent host mRNA transcription was not evaluated (referring to whether the effects of Zta on open chromatin produce changes in nascent transcription), but this subtlety is hard to convey in a figure legend, so we  are deleting that phrase.  The Frey paper examines a more complex system (not isolated Zta) and is best addressed under non-canonical host shutoff effects, as we do for the effects of Zta on chromatin (lines 292-295).

Line 148-150. This difference in the binding pocket provides additional evidence that gammaherpesvirus host shutoff factors target different host mRNAs. It is likely that there are subsets of host and viral genes that are targeted by both BGLF5 and SOX.  
Response:  We did not intend to imply otherwise.  We have edited this to try to clarify our stance on the issue.

Line 93, rendering them incompetent for translation, rather than rendering them for incompetent translation.
Response:  Thanks for catching this.  Our version was comically bad.

Line 146, consistent with the “for”, for is misplaced.
Response:  Actually this should read “four” as pointed out by reviewer 2 – sorry for the confusion.